# Preclinical Evaluation of TB/FLU-04L—An Intranasal Influenza Vector-Based Boost Vaccine against Tuberculosis

**DOI:** 10.3390/ijms24087439

**Published:** 2023-04-18

**Authors:** Anna-Polina Shurygina, Natalia Zabolotnykh, Tatiana Vinogradova, Berik Khairullin, Markhabat Kassenov, Ainur Nurpeisova, Gulbanu Sarsenbayeva, Abylai Sansyzbay, Kirill Vasilyev, Janna Buzitskaya, Andrey Egorov, Marina Stukova

**Affiliations:** 1Smorodintsev Research Institute of Influenza of the Ministry of Health of the Russian Federation, 197022 St. Petersburg, Russia; 2Saint-Petersburg State Research Institute of Phthisiopulmonology of the Ministry of Health of the Russian Federation, 191036 St. Petersburg, Russia; 3Research Institute for Biological Safety Problems, Gvardeiskiy 080409, Kazakhstangukacool@mail.ru (G.S.);

**Keywords:** *M. tuberculosis* vaccine, Ag85A, ESAT-6, influenza vector, mucosal immunization

## Abstract

Tuberculosis is a major global threat to human health. Since the widely used BCG vaccine is poorly effective in adults, there is a demand for the development of a new type of boost tuberculosis vaccine. We designed a novel intranasal tuberculosis vaccine candidate, TB/FLU-04L, which is based on an attenuated influenza A virus vector encoding two mycobacterium antigens, Ag85A and ESAT-6. As tuberculosis is an airborne disease, the ability to induce mucosal immunity is one of the potential advantages of influenza vectors. Sequences of ESAT-6 and Ag85A antigens were inserted into the NS1 open reading frame of the influenza A virus to replace the deleted carboxyl part of the NS1 protein. The vector expressing chimeric NS1 protein appeared to be genetically stable and replication-deficient in mice and non-human primates. Intranasal immunization of C57BL/6 mice or cynomolgus macaques with the TB/FLU-04L vaccine candidate induced *Mtb*-specific Th1 immune response. Single TB/FLU-04L immunization in mice showed commensurate levels of protection in comparison to BCG and significantly increased the protective effect of BCG when applied in a “prime-boost” scheme. Our findings show that intranasal immunization with the TB/FLU-04L vaccine, which carries two mycobacterium antigens, is safe, and induces a protective immune response against virulent *M. tuberculosis.*

## 1. Introduction

Tuberculosis (TB) is one of the most significant causes of morbidity and mortality among all infectious diseases. The World Health Organization (WHO) reported 6.4 million new cases of TB in 2021, resulting in 1.6 million deaths [1]. The evolution of multidrug- and extensively drug-resistant strains of *M. tuberculosis* and the high prevalence of TB-HIV co-infection, as well as the COVID-19 pandemic, have significantly complicated global TB control [2]. Current efforts evidently fall short in achieving the WHO’s strategy to eliminate TB by 2035, which includes a 95% reduction in TB deaths and a 90% reduction in TB incidence compared to 2015 [1].

Vaccination remains of high importance in the control of *M. tuberculosis* (*Mtb*) infection. The only current TB vaccine, *M. bovis* bacillus Calmette–Guerin (BCG), is reliably effective in the protection of infants against disseminated tuberculosis and meningitis, but its efficacy against lung TB is highly variable [3]. This determines the demand for developing alternative anti-TB vaccines. Since tuberculosis is an airborne disease, the intranasal route of vaccine administration might be preferable for future vaccines. In contrast to systemic immunization, the respiratory route provides activation of mucosal homing mechanisms [4] that enables Ag-specific memory T cells to preferentially come back to the site of vaccination, thus inducing strong systemic and mucosal immunity [5]. In addition, mucosal vaccines are needle-free, which makes them painless and limits the risks of pathogen transmission associated with contaminated needles [6].

In this context, replication-deficient viral vectors expressing particular *Mtb* antigens appear to be an attractive approach to mucosal vaccination against tuberculosis. To date, several recombinant viral vectors, including replication-deficient adenovirus [7,8,9] modified Vaccinia Ankara virus [9], Parainfluenza type 2 virus [10], and Vesicular Stomatitis Virus [11], have been tested for mucosal vaccination against *Mtb* and have demonstrated considerable efficiency. In this respect, live attenuated influenza vector-based vaccines have promising potential in TB protection, as we have shown earlier for an influenza vector expressing ESAT-6 [12,13]. The insert capacity of a new vector backbone allowed us to include a second protective Ag85A antigen to induce a broader TB-specific immune response. In contrast to ESAT-6 protein, Ag85A is expressed by conventional BCG vaccine, creating a possibility to perform “prime-boost” vaccination utilizing the BCG → influenza vector immunization scheme.

In the present study, we show the results of the preclinical evaluation of safety, immunogenicity, and protective efficacy of TB/FLU-04L—a replication-deficient intranasal influenza vector vaccine, expressing two *Mtb* antigens, ESAT-6 and Ag85A.

## 2. Results

### 2.1. TB/FLU-04L Construction and Characterization

A recombinant influenza vector FLU/ESAT-6_Ag85A expressing ESAT-6 and Ag85A antigens of *M. tuberculosis* from the influenza A/Puerto-Rico/8/34 virus NS1 open reading frame was constructed by the reverse genetics method as described previously [14]. Attenuation of the virus was achieved by truncation of the NS1 protein to N-terminal 106 amino acid residues (normal size is 230 aa) and replacement of its C-terminal part by foreign sequences derived from *M. tuberculosis* (Figure 1a). The resulting virus could not efficiently grow in interferon competent cells or mammals due to impaired functions of the chimeric NS1 protein, rendering it as a replication-deficient vector.

The vaccine virus was able to replicate up to 8.0 log_10_ of TCID_50_/mL in Vero cells (Figure 1b) and was genetically stable after 15 serial passages (Figure 1c). The expression of chimeric NS1_106_-ESAT-6 protein and Ag85A protein was confirmed by immunofluorescence analysis with specific anti-ESAT-6 and anti-Ag85A antibodies. As expected, the intracellular localization of the ESAT-6 antigen was predominantly nuclear due to its fusion with the truncated NS1 protein, whereas the Ag85A antigen was driven by the leader peptide and accumulated in the Golgi compartment (Figure 1d).

For further studies, the TB/FLU-04L vaccine candidate was produced in WHO-certified Vero cells under serum-free conditions following a multistep purification process and formulated as purified FLU/ESAT-6_Ag85A virus in a sucrose-phosphate-glutamate stabilizing buffer. The shape of viral particles was controlled by electron microscopy. Vaccine virions had intact structures and looked like round-shaped particles with surface glycoproteins protruding from the envelope (Figure 1e).

### 2.2. TB/FLU-04L Is Safe and Immunogenic in Mice

To confirm an attenuated phenotype of the influenza vector expressing *Mtb* antigens, the ability of TB/FLU-04L to replicate in mice was compared with that of the wild-type (wt) influenza virus A/PR/8/34. Groups of twelve or ten 7–8 weeks old female C57bl/6 mice were inoculated with TB/FLU-04L or wt A/PR8/34. Viral loads in nasal turbines and lungs were determined on days 3 and 5 post-inoculation. No signs of TB/FLU-04 replication were detected at any time point after immunization in contrast to the wt A/PR/8/34 virus that was isolated from lungs and nasal turbines of all animals at both time points and replicated up to 5.33 log_10_ TCID_50_/mL (Figure 2). These data indicate high attenuation of the TB/FLU-04L vaccine candidate virus in comparison with the wt A/PR/8/34 virus. Despite the lack of vaccine virus replication, single immunization with TB/FLU-04L caused moderate seroconversion against influenza antigen in five animals out of six (HAI assay; GMT = 8.98 (4–32)).

The percentages of CD4^+^ and CD8^+^ effector memory T-lymphocytes (T_em_), which produced any combination of IFNγ, IL2, and TNFα in response to in vitro stimulation with the recombinant Ag85A and ESAt-6 proteins, were evaluated by the means of flow cytometry three weeks after a single immunization with TB/FLU-04L (Figure 3).

Both spleen and lung-derived CD4^+^ cells from the vaccinated animals exhibited cytokine production by the T_em_ subpopulation in response to Ag85A as well as to ESAT-6 antigen stimulation. Ag85A-specific CD8^+^ cytokine-producing T_em_ cells were detected predominantly in the lungs of vaccinated animals, whilst ESAT-6-specific cells were present in both organs.

Substantial functional heterogeneity was observed in the T-cell response. Along with the formation of single-positive cytokine-producing cells, vaccination led to the appearance of polyfunctional double-positive (IFNγ^+^IL2^+^TNFα^−^, IFNγ^+^IL2^−^TNFα^+^) and triple-positive (IFNγ^+^IL2^+^TNFα^+^) subpopulations of both CD4^+^ and CD8^+^ T_em_ cells.

### 2.3. Safety and Immunogenicity of TB/FLU-04L in Macaca Fascicularis

We assessed the performance of TB/FLU-04L in *Macaca fascicularis*, focusing on its attenuation and immunogenicity. In contrast to small animals, macaque monkeys are likely to be a better animal model for evaluating the effects of intranasal immunization [15]. These non-human primates were shown to be a suitable model both for tuberculosis infection [16] and TB vaccine [17] studies.

Seven adult macaques (2.5–6 years old) were immunized intranasally (i.n.) with 0.5 mL TB/FLU-04L (7.5 log_10_ TCID_50_/animal) twice, with a three–week interval between vaccinations. Vaccination did not provoke any clinical manifestation in the macaques such as fever, weight loss (Appendix A), or respiratory symptoms, confirming the safety of the vaccine in the primate model.

Vaccine virus RNA was detected by PCR in the nasal swabs of two animals on day 2 and one animal on day 4 post-immunization (p.i.). The replication-deficient phenotype of the TB/FLU-04L was proven by the absence of virus isolation from the nasal swabs taken on days 2, 4, and 6 p.i. Relative to the pre-immunization baseline, intranasal immunization with TB/FLU-04L resulted in higher IL6 levels in nasal secretions collected 24 and 48 h after immunization (Appendix A).

For immunogenicity evaluation, blood samples were collected on days 0, 21, and 42. *Mtb* antigen-specific response was studied in vitro by stimulating PBMCs with Ag85A and ESAT-6 recombinant proteins and subsequently determining the IFNγ levels in the cell supernatants (Figure 4b). One animal (14.3%) had detectable IFNγ response after the first immunization; after the second immunization, five out of seven macaques (71.4%) had significant elevation of IFNγ levels in response to the stimulation with both *Mtb* antigens.

According to the results of hemagglutinin inhibition assay (HAI) in primates, a single TB/FLU-04L intranasal immunization caused the serum antibody response to the viral vaccine vector in 50% of animals (GMT = 6.56 (<4–32)), and after the second immunization, seroconversion was observed in 71.4% of animals (GMT = 64.05 (8−≥512)) (Figure 4a). There was a strong correlation between antibody response to the viral vector and the *Mtb* antigen-specific T-cell recall (Appendix A). Therefore, intranasal immunization of non-human primates with TB/FLU-04L was safe, provoked no virus shedding, and induced specific immune responses against both influenza vector and *Mtb* antigens. A better response was achieved after the second immunization.

### 2.4. Two Intranasal Immunizations with the Influenza Vector Protect from Intravenous M. tuberculosis Infection at a Level Comparable with M. bovis BCG

Following two i.n. immunizations with TB/FLU-04L, mice were challenged intravenously (i.v.) with a virulent *Mtb* strain—*M. tuberculosis* Erdman 6.0 log_10_ pfu/animal (10^6^ CFU). Summarized biometrics and histological and bacteriological data indicated that the double i.n. immunization of mice with the TB/FLU-04L vaccine delayed the development of experimental tuberculosis. In comparison to BCG, the TB/FLU-04L vaccine candidate showed a commensurate protective effect. Although the bacterial load in the lungs and spleen following the challenge was similar in both groups (Figure 5a,b), there were differences between the TB/FLU-04L and the BCG vaccines in the cellular composition of granulomas and the severity of lymphohistiocytic infiltration (Figure 5c).

Histological examination of the lungs of non-vaccinated animals revealed intensive, large inflammation foci which reduced the airiness of lung tissue by more than 30% of the sections. In 6 out of 9 mice (66.7%), lung lesions contained large numbers of degrading neutrophils and nuclear debris. In the TB/FLU-04L group, the lung histological picture was characterized by the prevalence of primarily lymphoid granulomas (66.7% versus 44.4% in the BCG group) and large lymphohistiocytic infiltrates (100% versus 88.9% in BCG group) (Figure 5c). In addition, the gross pathology score of lung injury in the TB/FLU-04L group was significantly lower than in the BCG group (2.46 ± 0.04 conventional units versus 2.71 ± 0.04 conventional units, *p* < 0.001).

### 2.5. Single Intranasal Boost Immunization with the TB/FLU-04L Influenza Vector Enhances the Protective Efficacy of BCG against Intravenous M. tuberculosis Challenge

The ability of the TB/FLU-04L influenza vector expressing two antigens of *M. tuberculosis* to enhance the protective immunity of BCG was investigated by comparing the bacillary load in lungs and spleen and pathological damage in the lungs of C57BL/6 mice after the i.v. *M. tuberculosis* Erdman challenge. The BCG and influenza vectors were administered using the prime-boost immunization scheme as shown in Figure 6a. Since memory T cells with high proliferative potential do not form until several weeks after the first immunization [18], a single booster immunization with the intranasal influenza vector was given four months after the BCG priming dose.

To determine whether the viral booster improved the prior BCG-induced protection, we challenged C57BL/6 mice with 10^6^ CFU of *M. tuberculosis* Erdman i.v. three weeks after the boost immunization. Four weeks later, bacterial burdens in the lungs and spleens were determined (Figure 6c,d). Compared to the non-vaccinated control group, a single subcutaneous (s.c.) dose of BCG decreased the CFU counts by 0.93 log_10_ CFU in the lungs (*p* < 0.001) and by 0.78 log_10_ CFU in the spleen (*p* < 0.0001). The level of protection induced by BCG was improved by one booster intranasal vaccination with the TB/FLU-04L influenza vector. In the case of heterologous prime-boost immunization, we observed a significant reduction in the bacillary load when compared to the BCG group (by 0.95 log_10_ CFU in the lungs (*p* < 0.0001) and by 0.43 log_10_ CFU in the spleen (*p* < 0.001)).

The evaluation of lung lesions confirmed the bacteriological data (Figure 6b). Visual inspection of lungs four weeks after the challenge revealed that the lungs of non-vaccinated mice had extensive gross pathology characterized by numerous large specific lesions (>10) distributed throughout the lung, and occasional tubercles with the areas of necrosis were observed in some of the control animals. Vaccination with BCG significantly reduced the gross pathology score (2.45 ± 0.08 vs. 2.88 ± 0.03 in the non-vaccinated control group, *p* < 0.001), but in the BCG prime → TB/FLU-04L boost group, the mean score for TB specific lesions in the lungs was statistically lower than that of the BCG-only vaccinated group (2.05 ± 0.08 vs. 2.45 ± 0.08, *p* < 0.001).

For the lung tissue of unvaccinated mice, histological examination identified confluent foci of specific infiltration that included macrophages, lymphocytes, epithelioid cells, and neutrophils **(**Figure 6b, histopathology not-vaccinated control 300×). Confluent foci of specific infiltration occupied more than 30% of the section area. Additionally, we observed epithelioid cell granulomas—large aggregates of epithelioid macrophages surrounded by lymphocytes (Figure 6b, histopathology not-vaccinated control 600×).

The histological changes in the lungs of prime-boost vaccinated animals were much less pronounced (Figure 6b**,** histopathology boost 300×). No confluent infiltration was shown, and the specific inflammation foci were small, lacked exudate, and had a perivascular or peribronchial localization. Alveoli and interalveolar septa infiltrations were presented by lymphocytes and macrophages and contained neither epithelioid cells nor neutrophils. All granulomas in the prime-boost immunized group of 12 cases were lymphoid (large lymphohistiocytic infiltrates) (Figure 6b, histopathology boost 600×).

In contrast, analysis of lung histological sections of the BCG vaccinated group revealed that confluent foci of specific infiltration were present in four out of twelve cases (*p* < 0.05), in single neutrophils in one case and in lymphoid granulomas in two out of twelve cases (*p* < 0.001); in eight out of twelve cases, granulomas consisted of large aggregates of epithelioid macrophages.

The change in the cellular composition of granulomas from predominantly epithelioid to predominantly lymphoid and increased lymphohistiocytic infiltration of the lungs, which was noted during the immunization with the BCG prime → TB/FLU-04L boost, indicating activation of the local immune response in the lungs due to the mucosal route of administration of TB/FLU-04L. The outcome of experimental tuberculosis was assessed for 5.5 months p.i. Consistent with pathological data, the survival rate of the prime-boost vaccinated animals was significantly higher (*p* < 0.001, Log-rank test) compared to the BCG-only group (Figure 6e).

### 2.6. Type 1 and Type 17 Cytokine Responses and Treg-Cell Frequency during M. tuberculosis Challenge (Spleen)

The protective efficacy of boost vaccination was accompanied by a better production of the Th1/Th17 cytokines by stimulated antigen-specific spleen cells that demonstrated clear Th1 polarization (Figure 7). Splenocytes from *M. tuberculosis*-infected mice from the BCG prime TB/FLU-04L boost group produced significantly higher amounts of IFNγ and IL17 after in vitro stimulation with BCG (10^5^ CFU) in comparison with spleen cells derived from control mice as well as BCG-only vaccinated non-boosted mice. The same tendency was found for IL1α, IL13, and IL22. The TNFα production by spleen cells of the infected BCG prime/TB/FLU-04L-boosted mice was similar to that of the non-boosted BCG group but significantly higher than that of the control mice. The ability of splenocytes to produce IL10 and IL6 did not differ significantly between all groups. The amounts of IL21, IL2, IL4, and IL5 following BCG stimulation were near or under the detection limit in spleen cells derived from mice of all groups but were detectable following the Concovalin A stimulation.

It is known that regulatory T cells (T_reg_) are implicated in TB progression in both experimental and human models [19]. We evaluated T_reg_-cell frequency in infected mice by analyzing Foxp3 expression in spleen CD4+ lymphocytes ex vivo. Figure 8 represents the populations of lung CD4+Foxp3+ cells in different groups studied. The mice in the prime-boost group had a lower percentage of T_reg_ cells within the CD4+ T-cell population in splenocytes compared with the control and BCG groups (*p* < 0.05). Therefore, we analyzed the ratio of spleen CD4+:CD4+Foxp3+ cells and found it to be significantly increased in the prime-boost vaccinated mice, which implies a better control of the infection than in the BCG-only vaccinated mice.

The results showed that the BCG priming coupled with the TB/FLU-04L boost down-regulated CD4+ CD25+ FoxP3+ T_regs_ and had the best protective effect among all studied groups. Boosting BCG with the TB/FLU-04L induced strong Th1-type immunity and down-regulated T_regs_, which was correlated with the best protection against *M. tuberculosis* infection in mice.

## 3. Discussion

There is evidence that intranasal mucosal vaccination provides remarkably better immune protection against pulmonary tuberculosis than systemic vaccination [20]. Intranasal vaccination can be mediated by vaccine vectors that are based on respiratory viruses. Recombinant viral vector vaccines have several advantages over protein-based or inactivated vaccines. Given their natural ability to deliver genetic information, viruses are an optimal choice for vaccination through gene delivery [21]. Even in the case of replication deficiency, infectious viral vectors can induce a full spectrum of humoral and cellular immune responses that initiate at mucosal surfaces. Importantly, the viral vectors may have self-adjuvating activities through the stimulation of innate immune systems [22] in a natural way, enabling the circumvention of mucosal tolerance.

During the last three decades, a wide variety of viral vectors belonging to different viral families has been developed [23]. In the majority of cases, emphasis was put on the creation of replication-deficient vectors with a high capacity to keep and express long transgenes. These are adenovirus vectors and several types of Vaccinia virus vectors that reached advanced stages of their clinical evaluation [24,25].

Nevertheless, the development of reverse genetics methods has presented a promising opportunity for utilizing RNA-containing viruses, such as influenza, in vaccine design. There are several general advantages of the influenza virus as a vaccine vector. The first one to mention is the availability of attenuated strains. A live attenuated influenza vaccine (LAIV), a type of influenza vaccine in the form of a nasal spray, has been developed in Russia and USA [26]. It is an attenuated vaccine administered intranasally. The LAIV based on cold-adapted strains is approved by FDA and EMA and licensed under the trade name *FluMist* in the United States and Canada and *Fluenz* in Europe. A similar vaccine of Russian origin is also licensed by BioDiem in India (2014) and China (2020). LAIV is shown to have certain advantages including an acceptable level of attenuation and lack of transmissibility or evidence of reversion, which implies its genetic stability [27]. Nasal antibody and cell-mediated immunity has also been demonstrated [28].

Additionally, another live influenza vaccine based on an engineered NS1 gene (DelNS1) was evaluated in Phase I and II clinical trials, showing a complete safety profile [29,30]. In contrast to the cold-adapted attenuated influenza vaccines, the DelNS1 vaccine is an example of a live replication-deficient vaccine that does not cause viral shedding from the respiratory tract of vaccinated individuals. This type of vaccine can be produced only in interferon-deficient Vero cells since the NS1 protein, an interferon antagonist, is deleted from the viral genome [31,32].

Another advantage of the influenza virus as a vector is the lack of chromosomal integration. In contrast to many other viral vectors, including DNA-containing adenoviruses or pox viruses which are theoretically able to integrate into the human genome [33], influenza viruses do not have a DNA phase in their replication cycle and therefore lack any potential mechanism of chromosomal integration.

Moreover, influenza viruses possess considerable antigenic variability that enables repeated vaccinations. One of the drawbacks of immunization with viral vectors may be the development of pre-existing or post-priming immune responses against the vaccine, preventing viral uptake during vaccination. A definitive feature of the influenza virus as a vector is that antibodies to different subtypes of influenza virus show little cross-reactivity. Therefore, by choosing influenza virus vectors belonging to different glycoprotein subtypes, pre-existing immunity in the host can be circumvented.

Furthermore, there are some specific advantages of vaccine vectors based on t influenza viruses with truncated NS1. The influenza virus encodes a non-structural NS1 protein, which can simultaneously antagonize the innate immune system of the host and enhance the translation of viral proteins [34]. Therefore, the deletion of this pathogenicity factor leads to the complete attenuation of the virus [31,32].

The concept of the current vaccine approach is based on the development of a live replication-deficient influenza virus vaccine that is highly attenuated through the alteration of the NS1 protein function. This effect is achieved by replacing the C-terminal part of the NS1 open reading frame with foreign sequences that encode protective antigens of other infectious diseases, such as *M. tuberculosis*. Importantly, the C-terminal deletion of NS1 is not only responsible for the attenuated phenotype but also results in stimulating an efficient immune response in the host due to enhanced induction of pro-inflammatory cytokines and interferons [22,35].

Previously, we generated several subtypes of attenuated recombinant influenza A viruses expressing the 6-kDa early secretory antigenic target protein (ESAT-6) of *M. tuberculosis* from the NS1 reading frame truncated to 124 amino acid residues. It should be mentioned that these vectors could replicate in mouse lungs and, thus, were not fully attenuated. The vectors were capable of inducing an *M. tuberculosis*-specific Th1 immune response in mice. Moreover, the intranasal immunization of mice and guinea pigs mediated protection against the mycobacterium challenge [12,13]. These data demonstrate that a protective vaccination against *Mtb* using a chimeric influenza virus as a vector can be very effective.

To achieve better safety parameters in the present study, we constructed a similar but more attenuated vector that expressed a much longer insert encoding two ESAT-6 as well as Ag85A antigens. Previous studies suggested that a bivalent composition has advantages in protective efficacy compared to its monovalent counterpart [36]. The new influenza vector vaccine TB/FLU-04L appeared to be replication-deficient in animals, presumably due to a shortened N-terminal part of NS1 protein (to 116 amino acid residues) and a substantially longer transgene inserted instead of the deleted C-terminal part of NS1. The vector was made using the Vero-adapted A/PR/8/34 strain that belonged to the old H1N1 subtype. We chose A/PR/8/34 because of its complete attenuation in humans, which resulted from more than 1000 passages in ferrets, mice, and eggs, and high reproduction activity in tissue culture and eggs. Besides this, A/PR/8/34 (H1N1)-derived reassortants were shown to be harmless as live intranasal vaccine strains in many clinical studies, even for people with chronic bronchopulmonary diseases [37].

The obtained vector demonstrated high replicative activity up to 8.0 lg TCID_50_/mL and was confirmed to be genetically stable after 15 consequent passages in Vero cells. The safety of TB/FLU-04L was proven in mice, guinea pigs, ferrets [38], and monkeys. There was no viral shedding detected in any animal model; the vaccine strain was completely attenuated in comparison with the wt A/PR/8/34 virus. Toxicology studies revealed no toxicological or allergological effects of the vaccine [38].

Even one intranasal immunization with TB/FLU-04L led to substantial systemic (spleen) and local (lungs) CD4^+^ and CD8^+^ T_em_ cell responses to both vaccine antigens in mice. Although antigen-specific T-cell response was characterized by a more pronounced frequency of IFNγ-producing cells, functional heterogeneity was observed. The detection of CD8^+^ responses using recombinant proteins as recall antigens most likely resulted from cross-priming, which was also reported in other studies [39,40].

In the context of efficient protection against tuberculosis infection, the pivotal role belongs to CD4^+^ Th1 cells, which are the main producers of IFNγ that, in synergism with TNFα, activate microbicidal effector mechanisms in macrophages [41]. The evidence showed that the main reason for the immune system failing to eliminate *Mtb* during acute infection, which causes persistent infection, could be a delay in CD4^+^ effector T-cell response caused by the ability of *Mtb* to hinder the migration of infected antigen-presenting cells from the lung to the draining lymph nodes [42]. Mucosal vaccine delivery is efficient at inducing T_em_ and T_cm_ at primary lymphoid sites [43,44]. It can also enhance the arrival of primed effector T cells— resident memory cells, T_rm_—that express pulmonary homing receptors to the lung, thus preventing the delay in the onset of adaptive immunity [45,46]. CD8^+^ T cells are also considered to be important in the mounting immune response to *Mtb* as they are involved in the lysis and apoptosis of infected cells and the subsequent killing of intracellular bacteria [47]. CD8^+^ T cells could simultaneously act as a source of Th1 cytokines [48]. It is known that the immune response to BCG lacks antigen-specific CD8^+^ T cells [49]. The T-cell response to both antigens was also proven in NHP. Notably, the correlation between antibody response to the viral vector and *Mtb*-specific T-cell recall was observed.

Protection studies in mice demonstrated that i.n. vaccination with TB/FLU-04L, even after systemic intravenous challenge with a high dose of mycobacteria, provided a substantial level of protection, comparable to that of the BCG vaccine alone. Moreover, the histopathological signs of tuberculosis infection in the lungs of animals immunized with TB/FLU 04 were less pronounced than in the BCG group.

We also tested the ability of TB/FLU 04 to increase the protective efficacy of BCG. In the experiment with BCG-prime TB/FLU-04 boost immunization followed by the i.v. *M. tuberculosis* challenge, we observed a significant reduction in the bacillary load in the lungs and spleen accompanied by a remarkable reduction in lung pathology when compared to the BCG group. The improvement of protection associated with the TB/FLU-04L BCG boosting is highly likely related to the ability of TB/FLU-04L to induce strong Th1-type immunity and down-regulate T_regs_.

Thus, our findings show that intranasal immunization with the TB/FLU-04L vaccine that expresses two mycobacterium antigens is safe and induces a protective immune response against virulent *M. tuberculosis.*

## 4. Materials and Methods

TB/FLU-04L construction, culturing conditions, and purification

Chimeric influenza virus construction A/Puerto-Rico/8/34_ NS_116__ESAT-6_Ag85A (TB/FLU-04L) was generated by reverse genetics. Briefly, the nucleotide sequence of the *ESAT-6* gene (291nts) was inserted after the 374 nucleotide position of the *NS1* gene open reading frame in the pHW2000 plasmid [50,51] followed by the sequence of a 2A autocleavage site [52], a modified mouse IgK-derived signal peptide [53], and 891 nucleotides coding Ag85A protein [14]. The plasmid pHW- NS_116__ESAT-6_Ag85A together with expression plasmids for PB1, PB2, PA, HA, NA, NP, and M proteins of the A/Puerto-Rico/8/34 (A/PR/8/34) virus were used for transfection of Vero cells by Nucleofection technique (Amaxa/Lonza, Basel, Switzerland) according to the manufacturer’s instructions. The supernatant was collected 96 h post-transfection, and 15 consecutive passages were performed in Vero cells (Figure 1a).

The vaccine candidate TB/FLU-04L was produced in WHO-certified Vero (ATCC) cells cultured under serum-free conditions. The harvest was purified by consequent clarification, concentration, and diafiltration steps and formulated in a sucrose-phosphate-glutamate stabilizing buffer (SPGN). The stabilizing buffer was given as a “mock” control in animal studies.

PCR

The molecular weight of the chimeric NS segment of the generated TB/FLU-04L virus was analyzed by qRT-PCR (SuperScript III Platinum One-Step System, Invitrogen, Cergy-Pontoise, France) using a sense primer Len -134 (5′-AGCAAAAGCAGGGTGACAAAG-3′) and an antisense primer NS843 (5′-CTCTTGTTCCACTTCAAAT-3′) [31]. The pHW–NS_116_-ESAT-6_Ag85A plasmid was used as a positive control.

Limited dilution assay

Vital titers were determined by limited dilution assay in Vero cells. Vero cells were seeded into 96-well plates at a density of 1.8 × 10^6^ per well. Cells grew for 24 h to reach a full monolayer, and then the cell culture medium was removed and serial 10-fold dilutions of samples in four repeats were applied to the wells (0.01 mL/well). Viral titers were calculated on the 5th day as 50% tissue infectious dose per ml using the Reed–Muench method [54].

Immunofluorescence

For immunofluorescence assay, Vero cells were infected with TB/FLU-04L at an m.o.i. of 2. GolgiPlug (BD Biosciences, San Jose, CA, USA) at a concentration of 1 μL/mL was added to inhibit cell protein transport 6 h p.i. In 10 h, the cells were fixed, permeabilized, and stained with a primary anti-ESAT-6 polyclonal rabbit antibody (Thermo Fisher Scientific, Waltham, MA, USA) or anti-Ag85A (Abcam, Cambridge, UK) polyclonal chicken antibody, followed by incubation with a secondary anti-rabbit antibody conjugated to Alexa Fluor 488 or an anti-chicken antibody conjugated to Alexa Fluor 555 (Thermo Fisher Scientific, Waltham, MA, USA). Signals were visualized by using a Carl Zeiss LSM 700 confocal microscope (Germany).

Electron microscopy

A copper grid (Sigma, USA) with a collodion cover was placed onto a drop of the sample for 60 s, and then the support was washed with distilled water and contrasted with 1.5% sodium salt of phosphotungstic acid, pH 7.1. The preparations were dried at room temperature and studied under the JEM-1011 electron microscope (JEOL, Tokyo, Japan) and supplied with a digital camera Morada (Olympus, Japan) at an instrumental magnification of 100,000–250,000.

Hemagglutination Inhibition Assay (HAI)

Sera were diluted 1: 4 with Receptor Destroying Enzyme (RDE; Denka Seiken, Tokyo, Japan) and incubated at 37 °C overnight (o.n.). Next, the enzyme was inactivated by heat treatment (56 °C for 30 min) and serial two-fold dilutions of sera were prepared in 96-well microtiter plates. 25 µL/well of the standardized antigen (4 hemagglutination units/25 µL) were added. After an incubation period of 1 h at room temperature (RT), 50 µL of 0.5% cRBC were added and plates were incubated at RT for 1 h.

Intracellular cytokine staining

For intracellular cytokine staining (ICS), single-cell suspensions were prepared from spleens and lungs. The right ventricle was perfused with 10 mL of ice-cold DPBS (Gibco, Gibco, Waltham, MA, USA) before the removal of the lungs and spleens. Mechanically dissociated spleen cells and previously digested lung tissue (45 min, 37 °C in Collagenase (Sigma, Switzerland) 0.5 mg/mL, DNAse I (Sigma, India) 10 μg/mL solution) were passaged through 70 μm cell stainer into RPMI 1640 (Gibco, Waltham, MA, USA) medium supplemented with 10% *v*/*v* FBS (Gibco, Waltham, MA, USA) and 1% penicillin-streptomycin solution (Gibco, Waltham, MA, USA). Cells were washed before and after erythrocyte lysis using Ammonium chloride lysing solution (0.15 M NH_4_Cl, 10 mM NaHCO3, 1 mM Na_2_EDTA) and seeded at a density of 1 × 10^6^ cells per well into flat-bottom 96-well tissue culture plates (Nunc, Roskilde, Denmark). The cells were stimulated with the recombinant ESAT-6 or Ag85A proteins (5 μg/mL, Novus Biologicals, Littleton, CO, USA) in a presence of anti-CD28 (BioLegend, San Diego, CA, USA) at 37 °C, 5% CO_2_, for 12 h. Medium alone and PMA (Sigma, China) were used as negative and positive controls, respectively. Following that, GolgiPlug (BD Biosciences, San Jose, CA, USA) reagent was added and cells were cultured for another 4 h. Following incubation, the cells were washed (300 g, 5 min) and stained with ZombiRed (BioLegend, San Diego, CA, USA) and surface markers CD8-PECy7 (BioLegend, San Diego, CA, USA), CD4 -PerCPCy5.5 (BD Biosciences, USA), CD 44-BV510 (BioLegend, San Diego, CA, USA), and CD62L-APCCy7 (BioLegend, San Diego, CA, USA). To reduce unspecific cell staining, the TrueStain reagent (BioLegend, San Diego, CA, USA) was used. Subsequently, the cells were washed and the intracellular staining with IFN-γ-FITC, IL-2-PE, and TNF-α (BioLegend, San Diego, CA, USA) was performed using the BD Biosciences Cytofix/Cytoperm kit (BD Biosciences, San Jose, CA, USA) according to the manufacturer’s instructions. T_reg_-cell frequency was evaluated using staining with ZombiRed (BioLegend, San Diego, CA, USA) and surface markers CD45-BV421 (BioLegend, San Diego, CA, USA), CD4-PerCPCy5.5 and CD8-PECy7 (BD Biosciences, San Jose, CA, USA) and CD 25-PE (BioLegend, San Diego, CA, USA) with subsequent intranuclear staining of FoxP3 by the means of True-Nuclear™ Transcription Factor Buffer Set (BioLegend, San Diego, CA, USA). Data were collected using a BD Canto II cytometer and analyzed in BD FACS Diva (BD Biosciences, San Jose, CA, USA) and/or Kaluza 2.1 software (Beckman Coulter, Bray, CA, USA). Gaiting strategies and representative plots are shown in Appendix A.

Lymphocyte stimulation assay

For specific stimulation of monkey PBMCs, recombinant *Mtb* proteins Ag85A and ESAT-6 (Novus Biologicals, Littleton, CO, USA) at a final concentration of 5μg/mL were used. Medium alone or 10 μg/mL Concanavalin A (Sigma, Saint Louis, MO, USA) served as a negative or positive control, respectively. Following 72 h of incubation, cell supernatants were collected and frozen at −70 °C. The level of IFN-γ in the PBMCs supernatants was measured with the BD OptEIA™ Monkey ELISA Set (BD Biosciences, San Jose, CA, USA) according to the manufacturer’s instructions. Absorbance was measured in BioTek Synergy™H1 (BioTek instruments, USA).

Mice splenocytes were cultured in the presence of medium only, ConA (2.5 μg/mL, Sigma, Saint Louis, MO, USA), or BCG (5 μg/mL). Levels of cytokines in the culture supernatants were measured after 72 h by using the LEGENDplex™ MU Th Panel (BioLegend, San Diego, CA, USA).

Animal Studies

All the animals were kept under controlled conditions and observed for any signs of disease. Experimental work was conducted in agreement with European and national directives for the protection of experimental animals and with approval from the competent local ethical committees. All experimental designs are given in the Appendix A.

Safety and immunogenicity in mice

For safety studies, groups of twelve or ten 7–8 weeks old female C57BL/6 mice were intranasally (i.n.) inoculated under slight ether anesthesia with 0.01 mL of TB/FLU-04L in a dose of 6.0 log_10_ TCID_50_/animal or A/PR/8/34 in a dose of 5.0 log_10_ TCID_50_/animal. Viral loads in 10% suspensions of nasal turbines and lungs were determined on days 3 and 5 post-inoculation by the limited dilutions assay in Vero cells.

For the immunogenicity study, 7–8 weeks old C57BL/6 mice were immunized once i.n. with 6.0 log_10_ TCID_50_/animal of TB/FLU-04L (6animals per group). Three weeks after the immunization, mice were sacrificed by cervical dislocation; blood samples for HAI and lungs and spleens for ICS were collected under sterile conditions. All experiments were performed in two independent repeats.

Safety and immunogenicity in monkeys

Safety and immunogenicity studies were conducted in *Macaca fascicularis* (Research Institute of Medical Primatology, Sochi). Seven adult macaques, aged 2.5–6 years, were immunized i.n. with 0.5 mL TB/FLU-04L (7.5 log_10_ TCID_50_/animal) spaced three weeks apart. A special aerosol device was used for vaccination. This device is registered in Russian Federation for delivery of live attenuated influenza vaccines and represents an insulin syringe with a nozzle, enabling aerosol application of the formulation.

During the whole study period, animals were monitored for any clinical signs through daily observations. Body weight and temperature measurements were carried out before the first immunization and on days 2, 4, 8, 21, and 42. In order to evaluate the vaccine virus shedding, nasal swabs were collected from macaques on days 2, 4, and 6 after immunizations and processed in a limited dilution assay in Vero cells. IL-6 concentrations were measured in nasal swabs collected before the first immunization and 24 and 48 h after it.

For immunogenicity evaluation, peripheral blood mononuclear cells (PBMCs) were obtained from 5 mL of heparinized blood collected from the macaques before each vaccination (day 0, day 21) and three weeks after the second vaccination (day 42). PBMCs were isolated by gradient centrifugation with 92% Ficoll-Paque^TM^ PLUS (Cytiva Sweden AB, Uppsala, Sweden). The cells were cryopreserved in FBS (Gibco, Waltham, MA, USA) with 10% DMSO (AppliChem, Germany) using step-wise freezing to −70 °C (Cryo1 °C Freezing Container, “Mr. Frosty”, Thermo Fisher Scientific, Waltham, MA, USA) and were transferred to liquid nitrogen the next day. After thawing, cells were used in lymphocyte stimulation assay.

Protection studies

For the evaluation of protection, mice were challenged intravenously (i.v.) with a virulent Erdman strain of *M. tuberculosis* (1 × 10^6^ CFU). The severity of experimental tuberculosis was assessed four weeks after the challenge by evaluating bacterial load in the lungs and spleen, calculating the lung damage score, and performing a histological evaluation of the inflammation process in the lungs. Based on the number and size of *Mtb*-specific lesions and the presence of necrosis areas in the lungs, gross pathological scores were graded from 1 to 4 conventional units (U) according to the following criteria: scanty small tubercles were estimated as 0.5 U, small tubercles (<5)—as 1.0 U, numerous small tubercles (>5)—as 1.5 U, occasional large tubercles—as 1.75 U, confluent small tubercles and occasional large tubercles—as 2.0 U, large tubercles (<10)—as 2.25 U, numerous large tubercles (>10)—as 2.5 U, numerous confluent tubercles—as 2.75 U, tubercles with areas of necrosis—as 3.0 U, and numerous large necrotic tubercles—as 4.0 U. In the case of lung maceration by serous liquid, the index was increased by 0.25–1.0 U, depending on the extent of the damage.

Statistical analyses

The GraphPad Prizm 9.0 software (GraphPad Software, Inc., USA) or R2 studio was employed for statistical analyses. All values were expressed as the mean ± standard deviation (SD) or standard error of the mean (SEM), as indicated. Groups were compared using the one-way or two-way ANOVA followed by Tukey’s multiple comparison test. The significance of the differences between survival times was evaluated using the Log-rank test. *p* < 0.05 was considered statistically significant.

## Figures and Tables

**Figure 1 ijms-24-07439-f001:**
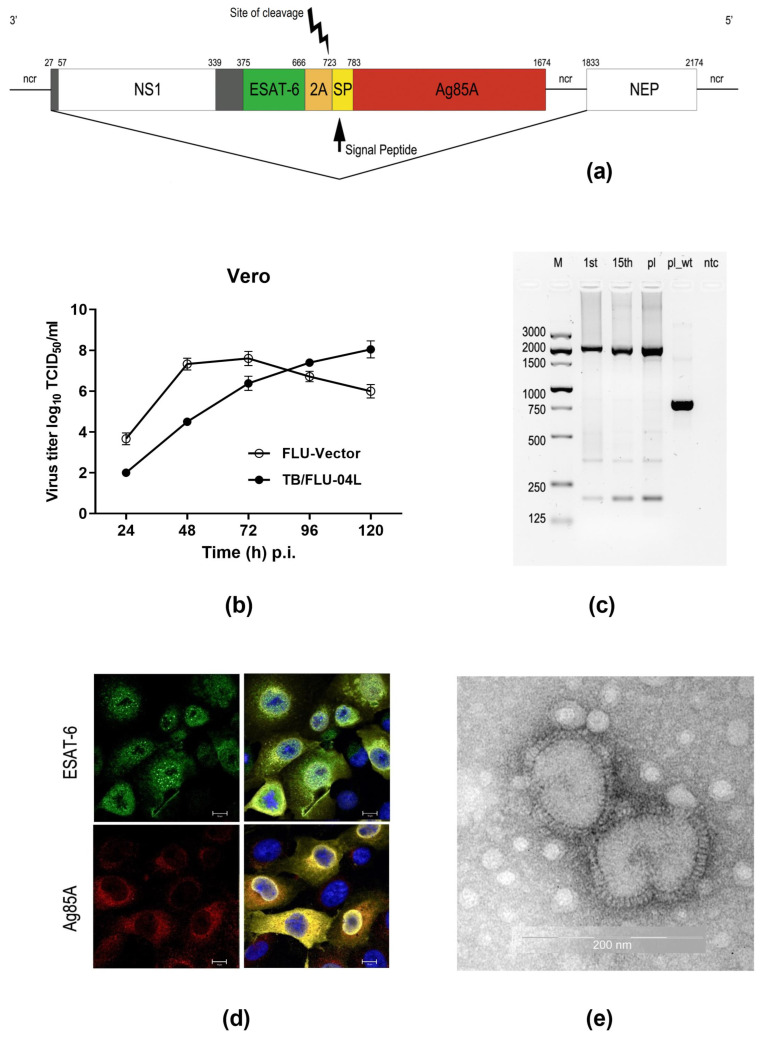
Construction and molecular characterization of the vaccine strain. (**a**) Schematic representation of the recombinant *NS* gene segment expressing antigens ESAT-6 and Ag85A of *M. tuberculosis*. The gray box (27–57) represents 12 aa that are shared between the NS1 and NEP proteins; the gray box (339–375) represents nucleotides encoding a random 12aa segment, fused to nucleotides encoding the ESAT-6 (green box) antigen followed by an autoproteolytic 2A cleavage site (orange box). The yellow box represents the nucleotide sequence of the modified mouse IgK-derived signal peptide (SP) followed by the Ag85A sequence (red box); ncr is a non-coding region. Nucleotide positions are indicated at the top. (**b**) Growth curve of TB/FLU-04L in Vero cells in comparison to an empty vector (FLU-vector). (**c**) Genetic stability of the chimeric *NS1* gene. RT-PCR analysis of the chimeric NS_116__ESAT-6_Ag85A segment in Flu/ESAT-6_Ag85A virus after 2 (P2) and 15 (P15) passages in Vero cells. Viral RNA was isolated from the supernatant of infected cells at 72 h p.i., and One-Step qRT-PCR was performed. The control used was pHW plasmid encoding NS_116__ESAT-6_Ag85A. M, Marker; pl, plasmid DNA; wt pl, pHW plasmid encoding NS1 gene segment from influenza A/PR8/34 wt virus. (**d**) Expression of ESAT-6, Ag85A, and NP proteins in infected Vero cells. Cells were infected with the Flu/ESAT-6_Ag85A virus (P5) at a multiplicity of infection (m.o.i.) of 2, then fixed and stained at 16 h p.i. using anti-ESAT-6 (green) or anti-Ag85A (red), anti-NP (yellow) antibodies, and DAPI (blue). Scale bars (white) correspond to 10 μm (**e**) Electron microphotograph of TB/FLU-04L viral particles following purification.

**Figure 2 ijms-24-07439-f002:**
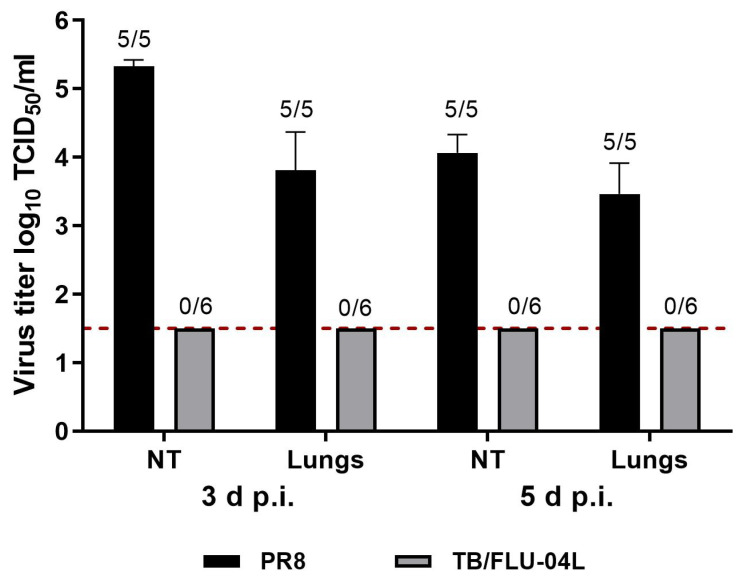
Virus shedding. Groups of ten or twelve 7–8 weeks old female C57bl/6 mice were inoculated with 0.01 mL of either TB/FLU-04L in a dose 6.0 log_10_ TCID_50_/animal or A/PR/8/34 in a dose 5.0 log_10_ TCID_50_/animal under slight ether anesthesia. Viral loads in 10% suspensions of nasal turbines and lungs were determined on days 3 and 5 post-inoculation in Vero cells. The virus titers are expressed as the mean log_10_ TCID_50_/mL ± SEM from 5–6 animals. The limit of virus detection was 1.5 log_10_ TCID_50_/mL (red dotted line). Tissues, where no virus was detected, were assigned the value of 1.5 log_10_ TCID_50_/mL for the calculation of the mean titer. The numbers placed above the bars reflect the ratio of animals with virus shedding to the total number of animals in the group. Mice with a lung virus load <1.5 log_10_ TCID_50_/mL were classified as not infected.

**Figure 3 ijms-24-07439-f003:**
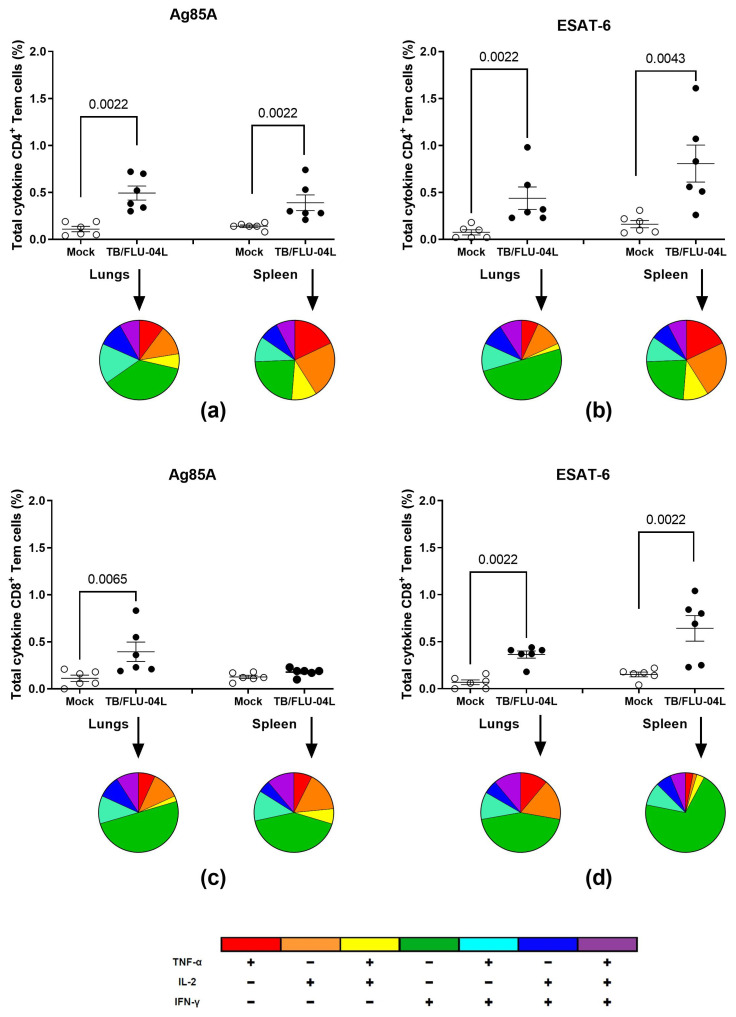
Antigen-specific immune response to TB/FLU-04L. Total frequency (%) of Ag85A (**a**,**c**) and ESAT-6-specific (**b**,**d**) cytokine-secreting CD4^+^ (**a**,**b**) and CD8+ (**c**,**d**) T_em_ cells were measured following a single TB/FLU-04L vaccination. Groups of six 7–8 weeks old female C57bl/6 mice were inoculated with 0.01 mL of DPBS (Mock) or TB/FLU-04L in a dose of 6.0 log_10_ TCID_50_/animal under slight ether anesthesia. Three weeks after the vaccination, single-cell suspensions prepared from the lungs and spleens of vaccinated mice were stimulated by the ESAT-6 and Ag85A recombinant proteins in vitro. The frequencies of IFN-γ-, TNF-α-, and IL-2-producing CD4^+^ and CD8+ T_em_ cells were measured by flow cytometry (ICS). Background staining from cells stimulated with medium alone has been subtracted. The data were considered statistically significant when *p* < 0.05 in the Mann–Whitney test.

**Figure 4 ijms-24-07439-f004:**
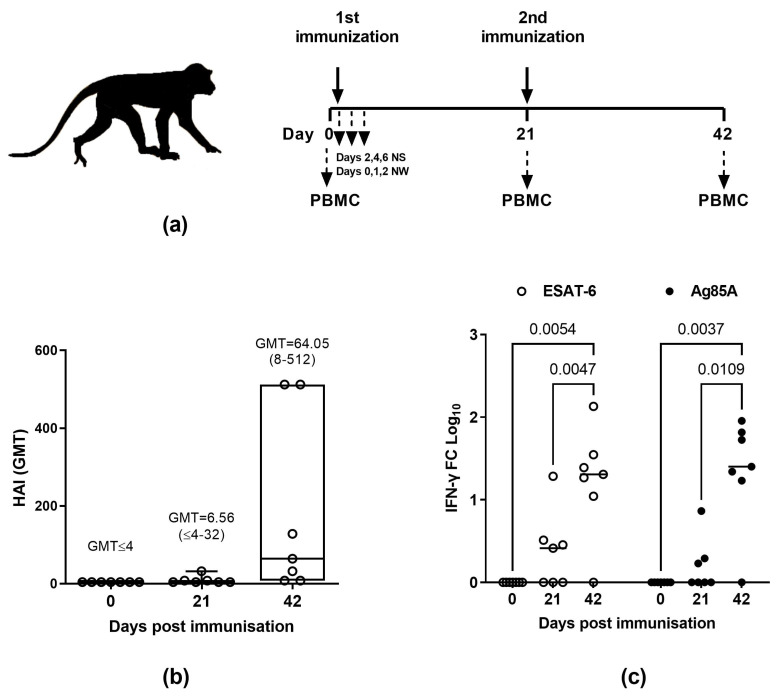
Immunogenicity of TB/FLU-04L in *Macaca fascicularis*. (**a**) Study timeline and sampling schedule. (**b**) Vector-specific antibody response in HAI and (**c**) antigen-specific IFNγ response (fold change [FC] log_10_) after the i.n. vaccination with TB/FLU-04L: seven adult male *Macaca fascicularis* were immunized twice with TB/FLU-04 (7.5 log_10_TCID_50_/animal), with a three-week interval between vaccinations. Blood samples for HAI assay and PBMCs were collected before each vaccination (day 0, day 21) and three weeks after the second vaccination (day 42). To assess the antigen-specific IFNγ response, PBMCs were stimulated in vitro by *M. tuberculosis* antigens (ESAT-6 or Ag85A 5μg/mL) for 72 h. Medium alone or Concanavalin A were used as a negative or positive control, respectively. Following incubation, cell supernatants were collected and IFNγ levels were determined by using the BD OptEIA™ Monkey ELISA Set. The data were considered statistically significant when *p* < 0.05 in the two-way ANOVA followed by Tukey’s multiple comparison test.

**Figure 5 ijms-24-07439-f005:**
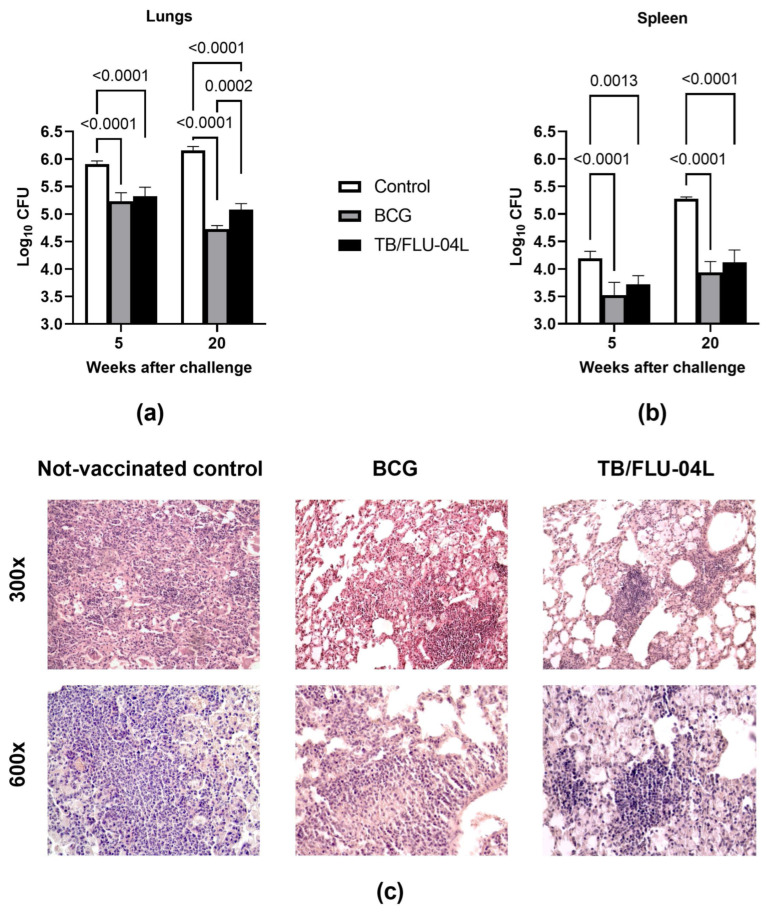
Protective immunity against virulent *M. tuberculosis* intravenous challenge in mice following intranasal vaccination with TB/FLU-04L. C57BL/6 mice received saline i.n. (naive), or 1 × 10^5^ BCG s.c., or TB/FLU-04L i.n. twice with three-week intervals. Three weeks after the second vaccination, mice were challenged by an i.v. injection of 10^6^ CFU of virulent *M. tuberculosis* Erdman strain. Graphs show means and SDs of CFU of *M. tuberculosis* in the lung (**a**) and spleen (**b**) 5 and 20 weeks after the challenge for groups of five mice that were either non-vaccinated control (white bars), BCG vaccinated (gray bars), or TB/FLU-04L vaccinated (black bars). The data were considered statistically significant at *p* < 0.05 in the mixed-model analysis with Tukey’s multiple comparisons test. (**c**) Development of lung pathology in three experimental groups of C57BL/6 mice five weeks after the i.v. challenge with *M. tuberculosis* Erdman (*n* = 10 per group). The photographs represent pathological lung tissue samples from the non-vaccinated control and mice vaccinated with BCG or TB/FLU-04L. H&E staining, magnification 300× (upper row), and 600× (lower row).

**Figure 6 ijms-24-07439-f006:**
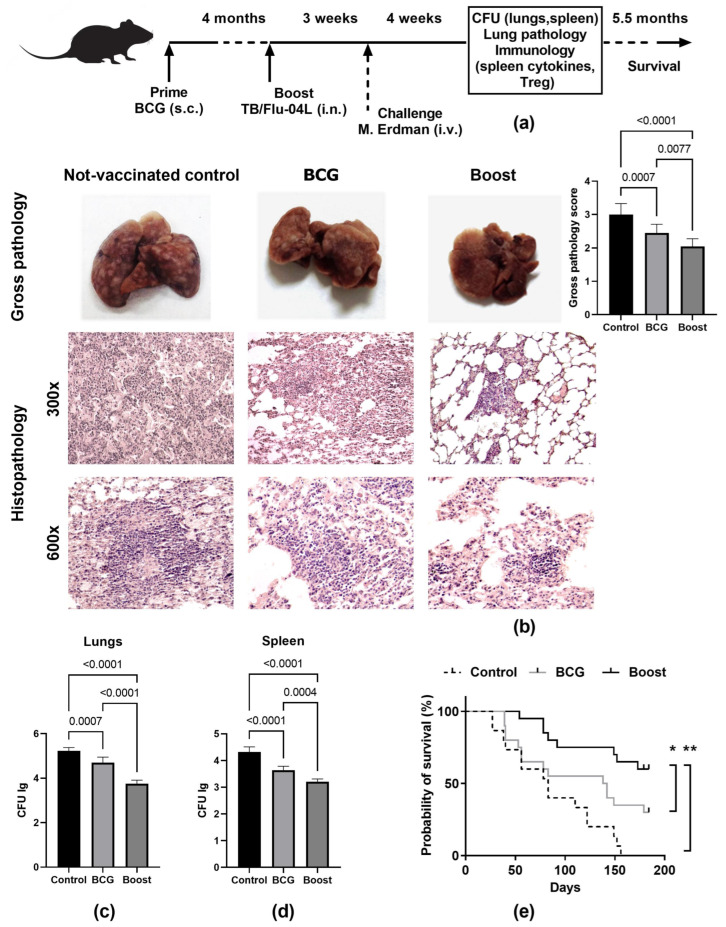
Protective efficacy of the BCG prime with TB/FLU-04L boost immunization against the i.v. *M. tuberculosis* challenge in C57BL/6 mice. (**a**) Study timeline and sampling schedule. C57BL/6 mice (40 animals per group) were immunized with a single s.c. dose of BCG (10^5^ CFU; BCG group) or with one dose of BCG administered s.c. followed by one intranasal immunization with the influenza vector expressing ESAT-6 and Ag85A proteins (10^6^ TCID_50,_ BCG prime/TB/FLU-04 boost group) four months apart. The control group was immunized with PBS only at the time of the prime and boost immunizations (non-vaccinated control group). Three weeks after the boost immunization, mice were challenged intravenously (i.v.) with the virulent Erdman strain of *M. tuberculosis* (1 × 10^6^ CFU). Four weeks after the challenge, the level of protection was measured by enumerating bacterial loads (CFU) in the lungs and spleens (6 animals per group) and evaluating gross pathological and histopathological changes in the lungs (10 animals per group). Cytokine secretion by spleen cells was measured at the time of sacrifice (4 mice per group). The survival of vaccinated and non-vaccinated C57BL/6 mice (15–20 animals per group) following the i.v. infection with *M. tuberculosis* Erdman was monitored over the 184-day observation period. (**b**) Tuberculosis lesions in the lungs of infected mice (*n* = 12 per group). Representative photographs of lungs, gross pathology scores, and histopathological appearances of the lung tissues stained with H&E. Magnification 300× (upper row) and 600× (lower row). (**c**,**d**) Bacterial growth in the lungs and spleens of vaccinated and control mice. The data were considered statistically significant when *p* < 0.05, as determined by the one-way ANOVA with Tukey’s multiple comparisons test. (**e**) The survival of C57BL/6 mice for 184 days post-infection. The significance of the differences between survival times was evaluated using the Log-rank test (* = *p* < 0.01, ** = *p* < 0.001).

**Figure 7 ijms-24-07439-f007:**
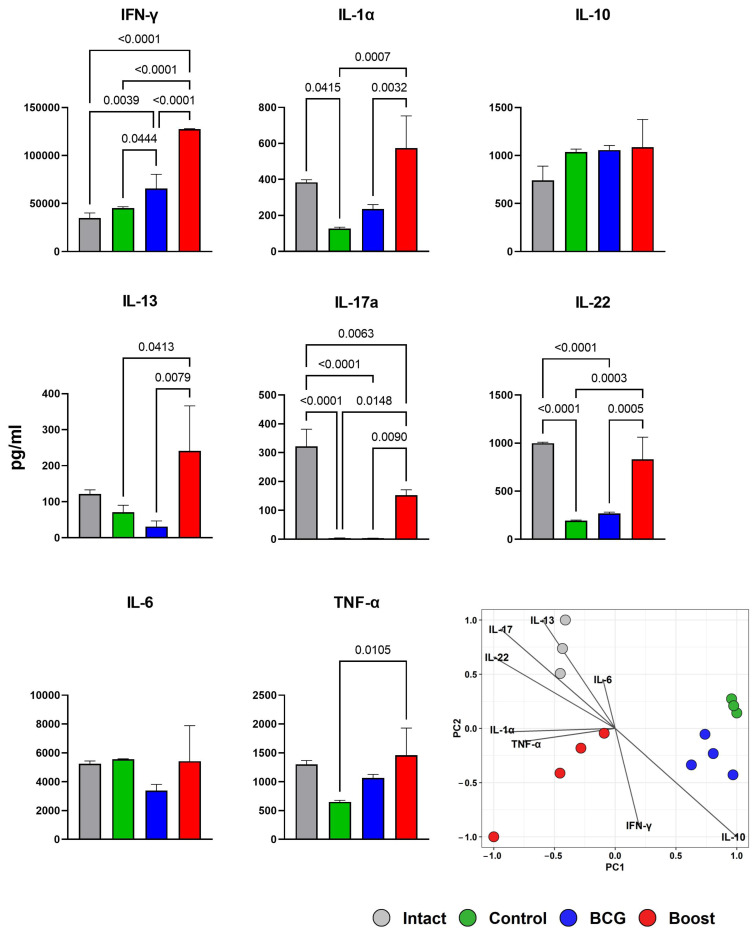
Spleen cell cytokine production in the BCG-vaccinated mice with or without TB/FLU-04L boost after *M. tuberculosis* infection. Four weeks after the i.v. challenge, spleen cells from intact (gray bars), control non-vaccinated mice (green bars), BCG-vaccinated mice (blue bars), or BCG-vaccinated/TB/FLU-04L-boosted mice (red bars) were cultured in the presence of medium only, ConA (2.5 μg/mL), or BCG (5 μg/mL). Levels of cytokines in the culture supernatants were measured after 72 h by using the multiplex kit (Biolegend). Values are expressed in pg/mL and represent group medians *±* SEMs of samples tested in duplicates. The data were considered statistically significant when *p* < 0.05 in one-way ANOVA with Tukey’s multiple comparisons test. Additionally, visualization of differences in cytokine response between the groups was performed using the Principal Component Analysis (PCA).

**Figure 8 ijms-24-07439-f008:**
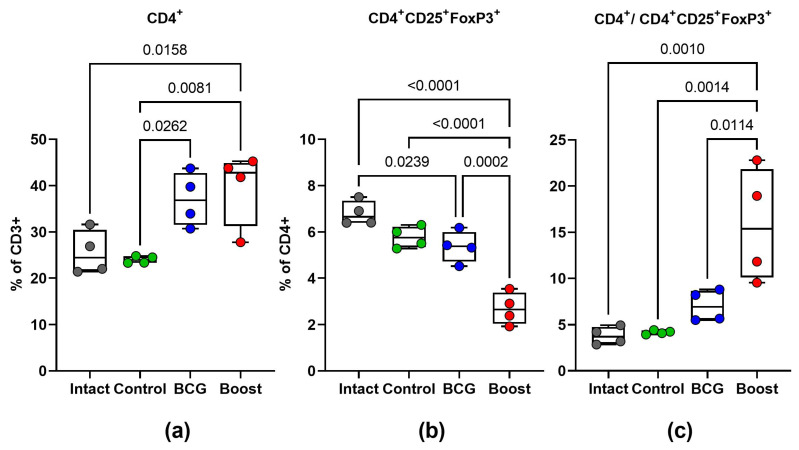
T_reg_-cell frequency during the *M. tuberculosis* challenge. Four weeks after the i.v. challenge, relative amounts of CD4^+^ T cells (**a**), CD4^+^CD25^+^FoxP3^+^ T_reg_ cells (**b**), and their ratios (**c**) were evaluated in spleen cells collected from intact mice (gray), non-vaccinated control mice (blue), BCG-vaccinated mice (green), or BCG-vaccinated/TB/FLU-04L-boosted mice (red). The data were considered statistically significant when *p* < 0.05 in one-way ANOVA with Tukey’s multiple comparisons test.

## Data Availability

The data presented in this study are available on reasonable request from the corresponding author.

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
