# Peer review of "Preclinical Evaluation of TB/FLU-04L—An Intranasal Influenza Vector-Based Boost Vaccine against Tuberculosis"

_ijms, 2023, doi:10.3390/ijms24087439_

Round 1
Reviewer 1 Report
Dear Editor
Thank you for your invitation to review the current manuscript.
Authors have designed a novel intranasal tuberculosis vaccine candidate—TB/FLU-04L, which is based on an attenuated influenza A virus vector encoding two mycobacterium antigens, Ag85A and ESAT-6. Then they tested its ability to induce Mtb-specific Th1 immune response by intranasal immunization of C57BL/6 mice or cynomolgus macaques with the TB/FLU-04L vaccine candidate. They found that single TB/FLU-04L immunization in mice showed commensurate levels of protection in comparison to BCG and significantly increased the protective effect of BCG when applied in a “prime-boost” scheme. They concluded that intranasal immunization with the TB/FLU-04L vaccine, which carries two mycobacterium antigens, is safe and induces a protective immune response against virulent M. tuberculosis.
Actually, this is a very interesting point and may have great impact on the field of TB protection.
My main and only comment is the study design is not clear and it should be presented with a diagram to show the groups of animals with their numbers and the process of vaccination.
Best regards,
Author Response
Dear reviewer, thank you for your kind feedback. Your remarks and recommendations allowed us to improve the manuscript by eliminating some minor inaccuracies and providing clarifications.
Response to Reviewer 1 Comments
Point 1. My main and only comment is the study design is not clear and it should be presented with a diagram to show the groups of animals with their numbers and the process of vaccination.
Response 1: All experimental designs were added to the supplementary file Tables S3-S6, which is referenced in the main text (lines 596-597).
The number of animals was corrected: five or six → ten or twelve (lines 113, 125, 599), 12→10 (line 229), 12→10 (line 298), 6→4 (line 299), 20-23→15-20 (line 300), 8→ 6 (line 605).
BCG dosage was corrected 106→105 (line 221).
The word “monkey” was inserted (line 581).
Reviewer 2 Report
To date, there is a need to develop a new generation of vaccines as the most effective immunoprophylactic means of combating tuberculosis. The strategy of heterologous vaccination finds the greatest support in the world, within the framework of which it is proposed to use the BCG vaccine or its improved analogues, or attenuated M. tuberculosis strains for priming the immune system, and subunit or vector vaccines containing protective proteins of mycobacteria for subsequent booster vaccinations. In the article under review, the authors present the results of preclinical evaluation of the replication-deficient intranasal influenza vector vaccine, expressing Ag85A and ESAT6 from M. tuberculosis.
In general, the article is of significant scientific interest for the TB community. Some minor issues to be addressed by the authors.
1. FACS-based pictures (Fig 3 and Fig 8) should be supported by the gating strategy in Supplement so as to give reader the possibility to evaluate the soundness of the data.
2. One would like also to see the photos used for gross pathology score counting (lines 238-240 and 267-271) at least in Supplement.
3. Using ESAT6 in the vaccine might interfere with further immunoscreening for TB (IGRA-tests and DIASKIN use ESAT6 as well).
As far as I can tell, the English of the article is sound enough. Though it might be improved by minor style editing.
Author Response
Dear reviewer, thank you for your kind feedback. Your remarks and recommendations allowed us to improve the manuscript by eliminating some minor inaccuracies and providing clarifications.
Response to Reviewer 2 Comments
Point 1. FACS-based pictures (Fig 3 and Fig 8) should be supported by the gating strategy in Supplement so as to give reader the possibility to evaluate the soundness of the data.
Response 1: Gaiting strategies for FACS-based pictures (Fig 3 and Fig 8) were added to the Supplementary Materials (Fig S3 and S4), which is indicated in the main text (lines 577-578). Methods were revised and the information concerning Treg staining was added (lines 572-576). Additionally, we made some minor changes in line 567 and the description of mice splenocyte stimulation and added cytokine measurements (lines 588-590).
Point 2. One would like also to see the photos used for gross pathology score counting (lines 238-240 and 267-271) at least in Supplement.
Response 2: Macropathology scores (lines 238-240 and 267-271) were calculated not by using photos but in real-time during the autopsy when a researcher examined all the surfaces of the lung lobes of mice according to the criteria given in lines 635-644. Photos cannot reflect all the existing foci of specific lesions and the degree of the lung’s maceration with serous fluid. Pictures of representative lungs (lines 267-271) are presented in Fig 6b.
Point 3. Using ESAT6 in the vaccine might interfere with further immunoscreening for TB (IGRA-tests and DIASKIN use ESAT6 as well).
Response 3: Thank you for the question. Indeed, there are concerns that vaccines whose protective mechanism is based on ESAT6- and/or CFP 10 antigens can potentially interfere with the latent TB diagnostic tests. In the context of the TB/FLU-04 vaccine application, this aspect will be studied in detail during further research in humans. However, the experience of using the MTBVAC vaccine indicated that the ELISPOT response elicited by MTBVAC following ESAT6 or CFP10 stimulation was below the cutoff established for tuberculosis infection [1]. In the future, if the protective vaccine based on the ESAT6 antigen is established, it would be reasonable to improve diagnostics of tuberculosis or reconsider the cutoffs of existing techniques that are used to identify individuals with a high risk of developing active TB [2].
- Spertini, F.; Audran, R.; Chakour, R.; Karoui, O.; Steiner-Monard, V.; Thierry, A.-C.; Mayor, C.E.; Rettby, N.; Jaton, K.; Vallotton, L.; et al. Safety of Human Immunisation with a Live-Attenuated Mycobacterium Tuberculosis Vaccine: A Randomised, Double-Blind, Controlled Phase I Trial. Lancet. Respir. Med. 2015, 3, 953–962, doi:10.1016/S2213-2600(15)00435-X.
- Aguilo, N.; Gonzalo-Asensio, J.; Alvarez-Arguedas, S.; Marinova, D.; Gomez, A.B.; Uranga, S.; Spallek, R.; Singh, M.; Audran, R.; Spertini, F.; et al. Reactogenicity to Major Tuberculosis Antigens Absent in BCG Is Linked to Improved Protection against Mycobacterium Tuberculosis. Nat. Commun. 2017, 8, 16085, doi:10.1038/ncomms16085.